# Challenges Faced by University of Limpopo Learner Nurses during Psychiatry Clinical Exposure: A Qualitative Study

L. S. Hlahla *, C. Ngoatle, M. N. Kgatla and E. M. Mathapo-Thobakgale

Department of Nursing Sciences, University of Limpopo, Polokwane 0700, South Africa
* Correspondence: sebolaisi.hlahla@ul.ac.za

**Abstract:** Clinical exposure of learner nurses to psychiatric hospitals is a requirement by the South African Nursing Council. Clinical experience helps learner nurses build cognitive and affective skills, cultural acculturation, and professional identity. The clinical placement also aids nursing learner nurses in making decisions regarding future career paths. The goal of psychiatric nursing practice is to enhance mental and physical health and improve the patient's quality of life and rehabilitation. A qualitative research approach was followed, and a descriptive, explorative, and contextual design was used in this study to explore the challenges faced by University of Limpopo learner nurses during psychiatry clinical exposure. The population included learner nurses from level two to level four who registered for psychiatric nursing science practice. Convenience sampling was adopted. Semi-structured interviews were used to collect data, and the data were analyzed using the Tesch open coding method. Measures to ensure trustworthiness were adhered to, and ethical considerations were observed. The findings of this study indicated that learner nurses go through challenges and discomfort in the form of mental health care users, clinical environment matters, and the attitude of clinical staff in the hospital. Proper preparation of the learner nurses and clinical areas can assist in reducing the challenges that learner nurses go through.

**Keywords:** challenges; learner nurses; psychiatry; clinical exposure

## 1. Introduction

The South African Nursing Council [1] specifies that in order to register as professional nurses, learner nurses must possess the necessary practical and clinical competencies. Therefore, nursing education institutions must provide learner nurses with opportunities for clinical learning through clinical placements before they graduate. The South African Nursing Council [1] further states that if a learner nurse does not receive 80% of clinical exposure hours in psychiatric nursing, that learner nurse will not be permitted to perform the final clinical assessment of a patient.

Clinical placement of learner nurses is an important part of nursing education and training. It gives learner nurses the chance to apply the knowledge and abilities they have learned in the classroom to real patients while under the supervision of professional nurses [2]. The development of their identities as future professional nurses depends on quality clinical exposure [3]. When the learner nurses are placed in the clinical area, they get an opportunity to apply theoretical knowledge learned in the classroom to practice [4]. Additionally, it helps them build their cognitive and affective skills, cultural acculturation, and professional identity. The clinical placement also aids learner nurses in making decisions regarding future career paths [5].

Nursing clinical training takes place in a complicated clinical learning environment that is influenced by a variety of factors, unlike education in a classroom. This setting offers learner nurses the chance to learn through experimentation and apply theoretical information to a range of required nursing skills that are important for psychiatric patient care. One of the key elements influencing the quality of clinical education is how learner

nurses are exposed to, and prepared for, the clinical environment [6]. Because an optimal clinical learning environment promotes the professional development of the learner nurses, a poor learning environment can be detrimental to their professional development. The unpredictability of the clinical training environment might cause challenges for learner nurses [6].

Psychiatric nursing is an area in nursing education and practice that brings specialized knowledge from nursing. Psychiatric nursing focuses on the mental health and wellbeing of individuals [7]. The goal of psychiatric nursing practice is to enhance mental and physical health and improve the patient's quality of life and rehabilitation. The main aim of psychiatric nursing is to assist patients to realize their problems and needs. Psychiatric and mental health nursing, in particular, is based on a trusted nurse–patient relationship [2].

While clinical placement gives the learner nurses a chance to build themselves professionally, it does, however, go without saying that learner nurses in clinical areas have challenges meeting their learning objectives during psychiatric clinical placements [8]. Learner nurses cannot naturally transfer knowledge from the classroom to a clinical setting; rather, their growth is influenced by their encounters and relationships with nurses and other professionals when they interact with actual patients [3].

Most learner nurses have their first encounter with psychiatric patients during their clinical placement in the hospital. Regardless of what they are taught in class, they have a tendency to maintain a belief that psychiatric patients are violent, dangerous, and hostile. These beliefs contribute to stress and anxiety in learner nurses regarding psychiatry clinical exposure. The stress and anxiety cause them to limit their interaction with the psychiatric patients [9].

A study conducted in Turkey indicated that students experience emotional challenges, such as fear, concern, anxiety, alienation, and loneliness, during mental health clinical education [10]. In South Arabia, it was discovered that poor preclinical exposure preparation makes students frustrated when they go for psychiatry clinical exposure [7]. A study conducted in Malawi showed that students have no confidence when they have to nurse psychiatric patients because of the stigma attached to mental illness [11]. Another study conducted in Gauteng Province, South Africa, reported that students do not receive the needed support from nurses in the hospital when they go for clinical exposure [12].

This study aimed to explore the challenges that learner nurses encounter when they are allocated for clinical exposure in psychiatric nursing institutions. This study is the first study at the University of Limpopo to explore the challenges facing learner nurses when they are placed in a psychiatric hospital for clinical exposure. The findings of this study will aid lecturers and nurses in hospital to assist students to adjust and perform their psychiatric clinical duties competently with minimal or no challenges.

## 2. Methods

A qualitative research approach was used in this study to help explain the challenges faced by University of Limpopo learner nurses during psychiatry clinical exposure. A descriptive, explorative, and contextual design was followed in this study. An explorative design was used to gather views from learner nurses regarding the challenges they face during psychiatry clinical exposure. The descriptive design was used by creating a diagram that showed the themes and the sub-themes, and the study was conducted in the context of the University of Limpopo.

### 2.1. Study Site

This study was conducted at the University of Limpopo, Mankweng, Limpopo Province. The University is approximately 36 km away from Polokwane city. Nursing programme is offered for four years at the University of Limpopo. When this study was conducted, the students were allocated for psychiatry clinical exposure from level 2 to level 4 of study. The students are taken through theoretical teaching and simulation before they can be taken to the hospital for psychiatric clinical exposure.

*2.2. Population and Sampling*

2.2.1. Population

The population for this study was University of Limpopo learner nurses enrolled for the psychiatric nursing science practice module. The total number of learner nurses who were eligible to be part of the study was 228 in 2022. That is 67 learner nurses at the 2nd level, 85 at the 3rd level, and 76 at the 4th level of study. All these learner nurses from the University of Limpopo were once allocated for psychiatric clinical exposure in mental health institutions so the researchers hoped they would provide the needed relevant information.

2.2.2. Sampling

The researchers used a non-probability sampling method to select participants for this study from the study population. The researchers chose readily available participants [13], and convenience sampling technique was used in this study.

Inclusion Criteria and Exclusion Criteria

Inclusion: All the learner nurses at the 2nd, 3rd, and 4th levels who were registered for psychiatric nursing science practice were included in the study. These learner nurses were included because they had been exposed to mental health/psychiatric institutions.

Exclusion: This study excluded all the learner nurses at the 2nd, 3rd, and 4th levels who registered for the psychiatric nursing science practice but were not available during data collection.

*2.3. Data Collection*

The primary author collected data at University of Limpopo. Private offices were used for data collection to ensure privacy. The interviews lasted from 45 min to 1 h. Data in this study were collected using semi-interviews, whereby a voice recorder was utilized to record the interviews, and the field notes were written down during the interview process. A consent form was signed during the briefing session before the interview, and the participants were informed that a tape recorder would be used in the interview process. An interview guide was used to direct the interviews, with the central question being, "What are the challenges you face as a learner nurse during psychiatry clinical exposure". Follow-up questions were then asked based on the participants' response. Data saturation was reached at participant number 22 where no further new information arose.

*2.4. Data Analysis*

Data were analysed using Tesch's eight-step open coding method as follows [14]: 1. The researchers read through the transcriptions and then continued to write ideas to obtain a sense of the whole picture. 2. Then, they started with the most exciting shortest interviews, considering the underlying meaning of information while writing views in the margins. 3. They further made a list of all topics coming from the transcript. 4. Then, they listed back to abbreviate the topics as codes and wrote the codes next to the relevant segment of the text. 5. They then identified descriptive phrasing for the topics and turned them into categories. 6. Abbreviations for each category were made, and researchers arranged the codes in alphabetical order. 7. Introductory analysis was made by collecting the data material belonging to each category in one place. 8. The researchers lastly summarized the themes and sub-themes developed, and then the raw data were sent to an independent coder who assisted with coding.

*2.5. Measures to Ensure Trustworthiness*

The following measures of trustworthiness were adhered to: credibility, dependability, transferability, confirmability, authenticity, and reliability. Credibility was ensured by the researchers spending more time in the field. Dependability was achieved by the researchers through the use of an independent coder. Transferability was adhered to by explaining the

methodology in depth. Confirmability was achieved through an audit trail. Authenticity was achieved through a deep, vivid description of the methodology used [14].

*2.6. Ethical Considerations*

Ethical clearance to conduct research was requested and received from the University of Limpopo Turfloop Research Ethics Committee (TREC/318/2022:UG-26 June 2022). Permission to conduct the study using University of Limpopo learner nurses was obtained from the Director of the School of Health Care Sciences and the Head of the Department of Nursing Science department. After receiving information about the study, the learner nurses voluntarily consented to participate in the study. A consent form was provided to learner nurses to sign before interviews. The researchers assured the learner nurses of confidentiality, privacy, and anonymity, and the principle of justice was also adhered to.

## 3. Results

*3.1. Demographic Data*

The number of learner nurses who participated in this study was twenty-two. There were seventeen female learner nurses who participated in this study, and there were five male learner nurses. The gender discrepancy is mainly because most learners who enroll for the profession at the University of Limpopo are females. The participants were from level two of study to level four. Data were collected until saturation was reached at participant 22. Table 1 below indicates the demographics of the learner nurses who participated in the study.

**Table 1.** Demographics.

|  | **Characteristics** | **Number** |
|---|---|---|
| **Gender** | Female | 17 |
|  | Male | 5 |
| **Level of study** | Level 2 | 5 |
|  | Level 3 | 8 |
|  | Level 4 | 9 |

*3.2. Themes and Sub-Themes*

The responses provided by the learner nurses during the process of collecting data led to the formation of themes and sub-themes, as tabulated below. Table 2 provides a presentation of themes and sub-themes that came out of the study.

**Table 2.** Themes and sub-themes.

| **Theme** | **Subtheme** |
|---|---|
| **1.Discomfort towards the mental health care users** | 1.1. Fear of mental health care users |
|  | 1.2. Uncertainty about learned psychiatry skills |
| **2.Clinical environment matters** | 2.1. Different clinical environment |
|  | 2.2. The limited time of exposure |
| **3.The attitude of clinical staff at the psychiatric hospital** | 3.1. Failure to supervise learner nurses |
|  | 3.2. Non-engagement of learner nurses in psychiatric procedures |

**Theme 1: Discomfort towards mental healthcare users**

This theme yielded two sub-themes, which were fear of mental healthcare users and uncertainty about their learned psychiatry skills.

**Sub-Theme 1.1. Fear of mental healthcare users**

The learner nurses reported that when they allocated to a psychiatric institution, they find themselves afraid of the mental healthcare users. The responses below are evidence:

Participant 3 mentioned the following:

*"When interacting with patients we get scared that the patients may hit us or something like that…"*

Participant 5 added, by saying the following:

*"Um, I had fear, to be honest. I already had stories about nurses being killed by healthcare users or being sexually assaulted. So I had that fear that what if I don't come back that day? What if I go to the wrong corner? And then a mental healthcare user comes after me? That was my greatest".*

Participant 8 said that because of the stories they heard about mental healthcare users, they felt unsafe.

*"Okay, when we are at Thabamoopo, we are having an emotional problem because we are not safe. After all, there are people who kill other people, and then meaning that we are not safe because we feel like those people will also kill us or rape other people. So that's why we are saying it affects me emotionally. Because sometimes I feel like I don't want to be close with them due to their threat to us that I have that a challenge….then the police or security must be available anytime or four to five police. Not only one".*

**Sub-Theme 1.2: Uncertainty about learned psychiatry skills**

The learner nurses reported that they were not comfortable being around mental healthcare users because even if they had learned about them in the classroom, they were not sure how to handle them. They voiced their opinions, as stated below, to confirm that they were not sure of the learned skills in the classroom.

Participant 18 reported that, at times, what they are taught in class does not correspond with what they find in the hospital.

*"…what we are taught in class and what we see at the psychiatric institution do not always correlate, for example in class, we are taught that each ward must have a therapeutic ward program; then when we get there some wards do not have those programs".*

Participant 20 gave a report that what they learn in the classroom does not seem through enough.

*"I refrain from interacting with mental health care users. I am not able to apply theory into practice, hence am saying we do not learn enough….We tend not to know how to interact with mental health care users. What we learn is not enough".*

Participant 11 added the following:

*"We do not have all the skills we will need to can use in the future to see if we can get this done so we are limited in the way we must learn".*

**Theme 2: Clinical environment matters**

The theme addressing clinical environment matters yielded two sub-themes, which were different clinical environment and limited time of exposure.

**Sub-Theme 2.2. Different clinical environment**

The learner nurses reported that they found the psychiatric nursing clinical environment to be rather different from the general nursing environment they were used to. They gave the accounts below to describe the psychiatric clinical environment.

Participant 11 found out that the way patient care is rendered in a psychiatric hospital is different to the patient care in a general hospital.

*"The challenge that I could say we encountered during psychiatric clinical allocation is that we don't get to do much patient care like the way we do in the general hospital. All we do is just to interact with the patients the whole day, and give them medications".*

Participant 3 added the following:

*"okay, I feel like it affects us when it comes to learning. Because, like I said that we don't get to do too much. And most of the time, we are just sitting outside and interacting with the patients. So, we are not exposed as to what is done during psych, and that it's a public or a government facility for treating people with mental illness".*

Participant 20 further said the following:

*"I expected to meet severely ill mental health care users. To be able to implement a lot of theory into practice but I found the patients mostly stable and communicating well".*

Participant 11 added the following:

*"Honestly, I didn't know what to expect. But I could say I did not expect the environment to be the way I saw it. So, I just thought maybe it was supposed to be like a normal hospital or it was going to be something else that of which is not what I saw in terms of arrangements, the way they sleep, how they eat and dress".*

**Sub-Theme 2.2: The limited time of exposure**

The learner nurses reported that they were only allocated to the psychiatric hospitals for a limited time and that this came as a challenge to their learning. They gave information through the quotes below.

Participant 6 reported that time they were given at the psychiatric hospital was limited.

*"uh, the challenges, one of the challenges is that we have uh one uh closer hospital of psychiatry around which limits the number of learner nurses to go there for clinical exposure whereby you find that you go there once or twice in the whole of the allocation so the exposure is very much limited".*

Participant 08 confirmed this, saying the following:

*"Oh, the time of exposure to the psychiatric hospital is limited and we are not exposed to all the wards. With the limited time of exposure, we end-up not having more knowledge about psychiatry".*

Participant 19 relayed

*"I was exposed to one unit, and therefore could not learn more about another user in different units…..I could not learn how to manage patients in other units because of limited time allocation. I was not able to learn enough without the assistance of the nurses".*

**Theme 3: The attitude of clinical staff**

This theme yielded two sub-themes regarding the attitude of the clinical staff in psychiatric institutions. The sub-themes were failure to supervise learner nurses and non-engagement of learner nurses in psychiatric procedures.

**Sub-Theme 3.1: Failure to supervise learner nurses**

The learner nurses reported that nurses in the psychiatric hospital did not supervise them. They spent most of their time alone without the supervision of professional nurses. They gave the below responses in talking about the lack of supervision from nurses:

Participant 16 said that they are not supervised in their nursing activities.

*"The first challenge is that the nurses at the psychiatric institution are rude to us. Another challenge is that we are not learning anything there; we just sit with patients the whole day until it is time to knock off…I expected that the nurses would involve us in the nursing care of the patients, to learn more about the psychiatric patients and their conditions".*

Participant 19 also added that

*"I was exposed to one unit and could not learn more about other users in different units. The staff does not give attention to the learner nurses…I could not learn how to manage patients in the other units, I was not able to learn enough without the assistance of the nurses".*

Participant 5 reported that the nurses did not look after them:

*"I wouldn't say so as much were normally. . .In my case, I feel like we don't get enough monitoring, or even really been getting monitored, to be honest. And the reality because most of the time you find that they're the staff nurses are in there where they normally sit, and we're outside with the patients like literally were the ones will you get to explain the patient alone without them explaining to us, like I said, before my challenges, we don't know this patient and their conditions. We just have to learn it by ourselves. Sometimes. Yes, we do check the files, of course. But if you're there with us, I feel like it'll be easier for us because why people write and really what they see or interact with, is much more different. So, we don't really get monitoring from the nurses who are in charge of us".*

**Sub-Theme 3.2: Non-engagement of learner nurses in psychiatric procedures**

The learner nurses reported that the nurses in the hospital were not engaging with them when they were performing psychiatric nursing procedures, and this led to the learner nurses feeling left out. The learner nurses gave the accounts below:

Participant 06 reported that

*"When we go to the hospital uhm, those Nurses, most of the time when we are there, they do not even give us the attention. . .".*

Participant 06 further added

*"We don't do anything, we do not even give medication, we don't even fill form or something, so they only call us maybe when they want us to take vital signs only when the doctors came there to check the patients".*

Participant 04 mentioned that

*"I also think of this one of the nurses not being able to, like, you know, they just do not teach us what we are supposed to do, like they do not".*

Participant 16 said this regarding non-engagement:

*"I expected that the nurses would involve us in the nursing care of the patients, to learn more about the psychiatric patients and their conditions".*

## 4. Discussion

This research aimed to explore the challenges that learner nurses faced during psychiatric clinical exposure. This study discovered that the challenges that the learner nurses go through include their discomfort towards mental healthcare users: the clinical environment they find themselves in and the attitude of clinical staff working in the psychiatric hospital. A study conducted in Iran confirmed that learner nurses face challenges in clinical areas when dealing with patients during clinical placement and when they have to interact with staff in the clinical environment [6].

The learner nurses reported that they were afraid of psychiatric patients. This could be because there are common mental illness stereotypes, including psychiatric patients being dangerous and aggressive [9]. There is a tendency of exaggerated crime rates and aggression of psychiatric patients, which makes learner nurses avoid them; this public discrimination against psychiatric patients may also be seen in the news and movies [15]. The learning objectives of learner nurses must focus on building the confidence of learner nurses and reducing their anxiety towards psychiatric patients [16].

The results of this study show that learner nurses have uncertainties about their learned psychiatric skills. This was supported by a study conducted in China by [15], who reported that it is common for learner nurses to feel nervous and afraid about taking care of psychiatric patients because some think they lack the necessary knowledge and abilities. The study further stated that the main reason for the learner nurses' worry and anxiety was their concern over how to handle patients with mental health problems. Due to fear, the learner nurses find themselves not knowing what to do when they are faced with psychiatric patients; they do not know how to act or interact with the individual [16].

The findings of the study indicated that learner nurses had issues with the psychiatric clinical environment. They found the psychiatric nursing environment to be different from the general nursing environment that they were used to. Psychiatric hospitals are specialized hospitals that are unique in that they are mostly in an isolated environment, with complicated patient conditions, heavy workloads, and high risks, necessitating higher professional skills and physical demands for nurses [17]. Proper orientation of the learner nurses to the setting can be useful in helping the learner nurses adapt to the environment [9].

The learner nurses reported that they are given a limited time in the psychiatric hospital; as such, they are unable to master the clinical skills related to psychiatric nursing. This was supported by [10], who said training and the type of clinical environment are significant factors influencing learner nurse competence in mental health clinical practice. The findings of their study reported that learner nurses failed to complete the practice and assessment phases of the nursing process and were unable to provide clients with holistic care because of the limited time exposure in the clinical area. The emphasis of the findings is that learner nurses need to be allocated a longer time in certain clinical areas for the learner nurses to be able to deliver competent nursing care.

The study findings show that the learner nurses met with the negative attitude of the staff at the psychiatric hospital. They were in the hospital without supervision, and they were not engaged in the procedures related to psychiatric nursing. When the learner nurses try to adjust to the psychiatric clinical setting smoothly, they face challenges. The adjustment of learner nurses is seriously hampered by feelings of abandonment and helplessness from the nursing staff. This is because the learner nurses felt like they were a burden on the nursing staff since they were rarely listened to and were neglected [18].

The learner nurses reported that they were not engaged in the psychiatric procedures by the nurses. The neglect of the learner nurses by the nursing staff in the clinical area is a constant struggle for learner nurses. This is observable when the nurses in the ward do not show interest in the learner nurses and choose not to supervise them. It is also observable when learner nurses are never given feedback to foster their learning in wards [17]. The learner nurses face challenges in the clinical areas where there are no preceptors when the learner nurses need them to assist them with practical nursing skills and the care of psychiatric patients. At times when the learner nurses go to clinical areas, nurses are not aware of their presence in the clinical setting, which leads to tensions in clinical settings. The majority of learner nurses thought the clinical environments were unwelcoming and they rarely felt welcomed [17,18].

## 5. Recommendations

Based on the findings of this study, it is evident that the learner nurses go through various challenges when they are allocated to a psychiatric hospital. Proper orientation of the learners, when they go to the psychiatric hospital, can be helpful to assist them to get used and adapt to the psychiatric hospital environment. The learning objectives of the learner nurses should be such that the fear of learner nurses is reduced, even before they go to the psychiatric hospital. Regular meetings between the university and psychiatric hospitals about the learning objectives of the learner nurses can be helpful in helping the nurses to understand why the learner nurses are in the hospital and what can be done to enhance their learning while they are in the hospital. The learner nurses need to be given enough time in the psychiatric nursing institution so that they learn and get used to the environment.

A study [16] recommended that nurse educators assist learner nurses in recognizing themselves, building awareness of the patient–nurse relationship, and good communication skills before their initial exposure to a psychiatric environment. The use of standardized patients can also be adopted during simulation to improve the learner nurses' relationship with patients. This means that the learner nurses will learn how to approach patients suffering from mental illness, and they will also learn how to communicate with them in preparation for the clinical environment [19]. Thus, the simulation of psychiatric skills for

learner nurses before clinical exposure should be comprehensive so that the learner nurses are ready emotionally and physically to work in a psychiatric hospital.

## 6. Limitations of the Study

The study was conducted on learner nurses at the University of Limpopo. Therefore, the results cannot be generalized to other learner nurses at other universities.

## 7. Conclusions

The findings of this study show that the learner nurses go through various challenges during psychiatric clinical exposure. The challenges they face range from discomfort towards mental healthcare users, which include fear of mental healthcare users to uncertainty about the learned psychiatric skills. Clinical environment matters were also raised as a challenge. Learner nurses found the psychiatric institution to be a different clinical environment from what they had been exposed to, and they were only exposed for a limited time. The other challenge was the attitude of clinical staff towards the learner nurses where the professional nurses failed to supervise the learner nurses and were not involving the learner nurses in psychiatric activities. The said challenges can be addressed through proper preparations before the learner nurses can be taken to the clinical institutions. Learner nurses need to be thoroughly prepared before they are exposed to the clinical areas. The hospitals where the learners are allocated must also be prepared to receive learner nurses so that learning continues in the institutions without fail.

**Author Contributions:** Conceptualization, L.S.H. and C.N.; methodology, M.N.K.; validation, C.N.; formal analysis E.M.M.-T.; investigation, L.S.H. and E.M.M.-T.; resources, C.N. and M.N.K.; data curation, L.S.H.; writing—original draft preparation, L.S.H.; writing—review and editing, L.S.H., C.N., M.N.K. and E.M.M.-T; visualization, M.N.K.; supervision, E.M.M.-T. and L.S.H. All authors have read and agreed to the published version of the manuscript.

**Funding:** This research received no external funding.

**Institutional Review Board Statement:** The study was conducted by the Declaration of Helsinki and approved by the Institutional Review Board (or Ethics Committee) of the Turfloop Research Ethics Committee (TREC/305/2018:PG, University of Limpopo).

**Informed Consent Statement:** Informed consent was obtained from all learner nurses involved in the study.

**Data Availability Statement:** The data used backing the findings of this study are available from the corresponding author upon request.

**Public Involvement Statement:** Participants in the study were the learner nurses from the University of Limpopo. Consent was obtained from all participants to use their personal stories as data.

**Guidelines and Standards Statement:** This manuscript was drafted against the Consolidated criteria for reporting qualitative research (COREQ) for Qualitative research.

**Acknowledgments:** The authors acknowledge all the students who took part in this study.

**Conflicts of Interest:** The authors declare no conflicts of interest.

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
