# Peer review of "Challenges Faced by University of Limpopo Learner Nurses during Psychiatry Clinical Exposure: A Qualitative Study"

_nursrep, doi:10.3390/nursrep14010014_

Round 1

Reviewer 1 Report

Comments and Suggestions for Authors

Dear Author,
the study is of interest for the preparation of nursing students.
Some fixes are recommended.
1. Title: Enter the type of study in the title
2. Introduction: I suggest further detailing the concept how the internship can be important to the student's professional identity.
3. Methods:
    - 2.1. Study site: The suggestion is to explain the didactic programming of the course in preparation for internships in psychiatry. How do you prepare students for internships in specialized areas such as psychiatry?
     - 2.2.1. Population: Who are the 2nd, 3rd and 4th level students. Does it mean the year of the course?
     2.2.1. Population: I recommend better explaining the population. All 228 students had internship experience in psychiatry. Did the entire population have a chance to be enrolled in the study? If so, why did only 22 students participate? (Table 1) Why was there not more significant participation? I recommend being clearer.
      - Inclusion Criteria and exclusion criteria: Better explain the inclusion criteria. It is unclear what the R174 and R425 nursing program is. What teaching content does the R174 and R425 program have?

4. Discussion: I suggest that the limitations of the study be included.

5. Recommendations: Too general. In light of: 1. the literature review; 2. your University I suggest evaluating simulation in psychiatry and evaluating a mentorship model in psychiatry internship that improves the clinical learning environment.

Author Response

Thank you for valuable

Reviewer 2 Report

Comments and Suggestions for Authors

Start by congratulating them for their work. It is considered to be an acceptable qualitative study; they present and define the study problem and its purposes very well.

The deficiency that was found is the number of participants, although it is possible to do the study with the number of patients included, however 22 seems to me to be a very low number for the size of the study population. Consider at the same time that the work would have been more productive incorporating more variables, they have only taken into account sex. I would advise taking age into account, dividing the participants based on the course, to try to establish what factors may influence these results. Would it be possible to reconsider it?

For the rest, the discussion is consistent and is compared with similar studies and the conclusions are consistent with the objectives and results.

Author Response

see attatchment below

Reviewer 3 Report

Comments and Suggestions for Authors

Dear authors,

The choice of the subject is very good and interesting, however, several points need to be improved in each of the sections of the article. Ιn addition, editing in English is required. The plagiarism report is attached for additional modifications (23%).

In detail, my comments are as follows:

Title

- Capitalize the first letter of every word in the title.

- learner nurses: Do you mean undergraduate students? Ιf so, perhaps you should consider replacing this term

Introduction

Τhe introduction could be improved on several points. The first few paragraphs essentially deal with the same issues in different words, while on the contrary, the issues concerning psychiatry, although it is the central theme of the analysis, are not sufficiently developed. Only one paragraph has been written on this subject, and in this paragraph only the subject of psychiatric nursing is mentioned. But why was the study of this particular clinical practice chosen? What is its specificity? what is the difficulty; what is the importance? For example, there is fear and stigma towards psychiatric disorders. Ιt is important to justify this choice, but also to mention the possible applications of the results in clinical practice and the benefits in general. Τhe organization of clinical training at the University should also be mentioned (from which year clinical training starts and in which year it takes place in psychiatric departments).

- line 27: “learner nurse nurses..” the word nurse is duplicated. Please correct.

- lines 29, 32, 71 and 76: Please correct the double space.

- lines 32-40: the word learner is repeated in every line. Please correct the syntax

- line 36: “nursing learner nurses”. Please correct.

- line 39: what is the symbol 0?

- lines 52-54: the wording of the sentence should be corrected to make the meaning clear

Methods

Τhe interview process should be described in more detail; by whom it was conducted, in which place, whether there was privacy and how long it lasted. In addition, it is required to document if the voice recorder is included in the student consent form.

- line 85: Put a full stop.

- line 94: Please correct the double space.

- line 100 & 107: : References must be numbered in the text.

- lines 124 & 143: Capitalize the first letter after full stop.

Results

Τhe themes and subthemes should be explained and properly analyzed. it is not enough to simply quote from the participants' respective speeches.

- Τables should follow the formatting of the text (font, etc.) and should not have vertical lines.

Discussion

- Possible peculiarities in clinical practice resulting from differences between countries should be described.

- Limitations of the study should be added.

Comments on the Quality of English Language

Extensive English editing is required.

Author Response

see attachment below

Round 2

Reviewer 3 Report

Comments and Suggestions for Authors

Dear authors,

Thank you for the modified version of the manuscript. Please capitalize the first letter of EVERY word in the title and check line 35, it is written "leaner nurses". .

Comments on the Quality of English Language

Minor editing in English is required.

Author Response

refer to the document attached below
